# MiR-218-5p/EGFR Signaling in Arsenic-Induced Carcinogenesis

**DOI:** 10.3390/cancers15041204

**Published:** 2023-02-14

**Authors:** Ranakul Islam, Lei Zhao, Xiujuan Zhang, Ling-Zhi Liu

**Affiliations:** Department of Medical Oncology, Sidney Kimmel Cancer Center, Thomas Jefferson University, Philadelphia, PA 19107, USA

**Keywords:** miR-218-5p, EGFR, arsenic, lung cancer, carcinogenesis

## Abstract

**Simple Summary:**

EGFR upregulation plays an important role in lung cancer as a well-established target for lung cancer therapy. However, the role and mechanism of EGFR upregulation due to chronic arsenic exposure remain to be elucidated. Here, we demonstrated that miR-218-5p was dramatically downregulated in arsenic-induced transformed (As-T) cells. It served as a tumor suppressor to suppress cell proliferation, migration, colony formation, and tube formation, and inhibit tumor growth and angiogenesis by directly targeting EGFR. Our results suggest that the 218-5p/EGFR signaling pathway may be a potential therapeutic target for the treatment of lung cancer induced by chronic arsenic exposure.

**Abstract:**

Background: Arsenic is a well-known carcinogen inducing lung, skin, bladder, and liver cancer. Abnormal epidermal growth factor receptor (EGFR) expression is common in lung cancer; it is involved in cancer initiation, development, metastasis, and treatment resistance. However, the underlying mechanism for arsenic-inducing EGFR upregulation remains unclear. Methods: RT-PCR and immunoblotting assays were used to detect the levels of miR-218-5p and EGFR expression. The Luciferase assay was used to test the transcriptional activity of EGFR mediated by miR-218-5p. Cell proliferation, colony formation, wound healing, migration assays, tube formation assays, and tumor growth assays were used to study the function of miR-218-5p/EGFR signaling. Results: EGFR and miR-218-5p were dramatically upregulated and downregulated in arsenic-induced transformed (As-T) cells, respectively. MiR-218-5p acted as a tumor suppressor to inhibit cell proliferation, migration, colony formation, tube formation, tumor growth, and angiogenesis. Furthermore, miR-218-5p directly targeted EGFR by binding to its 3′-untranslated region (UTR). Finally, miR-218-5p exerted its antitumor effect by inhibiting its direct target, EGFR. Conclusion: Our study highlights the vital role of the miR-218-5p/EGFR signaling pathway in arsenic-induced carcinogenesis and angiogenesis, which may be helpful for the treatment of lung cancer induced by chronic arsenic exposure in the future.

## 1. Introduction

Arsenic (As) is ubiquitous in the environment and is considered a toxic metal and a serious global health risk factor. Essentially, arsenic gradually leaks from the Earth’s crust and bedrock into drinking water [1]. According to the International Agency of Research on Cancer (IARC), arsenic is a Group 1 human carcinogen [2,3,4]. In the USA alone, nearly 3.7 million people drink water from private wells where arsenic contamination exceeds the USEPA standard of 10 ppb [5]. Humans are exposed to arsenic through arsenic-containing drinking water, food, air, or dust. Chronic exposure to dietary arsenic is linked to lung, skin, bladder, and liver cancer [6,7,8,9]. Arsenic-contaminated drinking water has been associated with increased death due to both noncancerous causes and cancers in Bangladesh [4,10]. Both acute and chronic arsenic toxicities generate harmful effects in multiple organs and tissues, such as hyperkeratosis, change in skin pigmentation, cardiovascular diseases, and developmental and cognitive impairments.

The epidermal growth factor receptor (EGFR), a tyrosine kinase receptor of the ErbB/HER oncogene, is among the cancer-related targets [11]. EGFR mutation or overexpression has an essential role in tumorigenesis in different types of cancers, including non-small-cell lung cancer (NSCLC) [12], breast cancer [13], colorectal cancer [14], and gastric carcinoma [15]. During the formation of a malignancy, EGFR plays a vital role in enhanced cell proliferation, angiogenesis, and metastatic spread [16,17]. Arsenic activates the EGFR pathway [18,19,20] and hampers DNA mismatch repair by promoting EGFR expression [21]. There is a positive correlation between arsenic concentrations in human toenails and the amount of EGFR protein present in NSCLC tumor samples [22]. It has been reported that arsenic exposure increases the EGFR ligand EGF in the lung and activates EGFR phosphorylation [22]. However, the underlying mechanism for arsenic-inducing EGFR upregulation remains unclear.

MicroRNAs (miRNAs) are a class of small noncoding RNAs that have been identified as a new kind of gene expression regulator by targeting the 3′-untranslated region (UTR) of mRNAs for translational downregulation or degradation [4,23,24,25]. Increasing evidence has demonstrated that miRNAs are involved in a wide range of biological processes, including cell proliferation, apoptosis, and migration [26,27,28]. Regarding cancers, miRNAs have been found to play essential roles as either tumor suppressors or oncogenes according to their expression levels and the involved downstream targets [4,29,30]. Recently, miR-218-5p has been shown to be downregulated, acting as a tumor suppressor in many human cancers, such as bladder cancer, hepatocellular carcinoma, gastric cancer, oral cancer, lung cancer, renal cell carcinoma, and glioblastoma [31,32,33,34,35,36,37,38]. Although recent studies showed that EGFR is one of the targets of miR-218-5p in NSCLC and pterygium epithelial cells [33,39], the function of miR-218/EGFR signaling in As-induced carcinogenesis and angiogenesis remains to be investigated.

In the present study, we aim to study the following: (1) whether the expression level of miR-218-5p is downregulated, and whether EGFR and its downstream targets are upregulated in As-induced transformed (As-T) cells; (2) the role of miR-218 in As-T cell-induced carcinogenesis and angiogenesis; (3) whether EGFR is a direct target of miR-218-5p; (4) whether miR-128-5p serves as a tumor suppressor through targeting EGFR. These findings will provide new insights into the molecular mechanism of As-induced carcinogenesis and angiogenesis, as well as a potential new therapeutic target in the future.

## 2. Materials and Methods

### 2.1. Cells, siRNAs, Plasmids, and Reagents

Human lung immortalized epithelial BEAS-2B (B2B) cells were purchased from ATCC (Manassas, VA, USA), and sodium arsenic (NaAsO_2_) was obtained from Sigma-Aldrich (St. Louis, MO, USA). We used 1-μM NaAsO_2_ to treat B2B cells for 24 weeks to establish arsenic-induced transformed (As-T) cells according to our previous publications [40,41]. In addition, 1-μM chromium was used to treat B2B cells to get Cr(VI)-induced transformed (Cr-T) cells [42]. For control, we used passage-matched B2B cells without As or Cr exposure. Skin cancer cells A431 were also purchased from ATCC (Manassas, VA, USA). A431, B2B, As-T, and Cr-T cells were cultured in complete media of Dulbecco’s modified Eagle’s medium with 10% fetal bovine serum (FBS) and 1% Penicillin and Streptomycin (DMEM; Invitrogen, Carlsbad, CA, USA). The cells were incubated with 5% CO_2_ at 37 °C. Human umbilical vein endothelial cells (HUVEC, purchased from ATCC, Manassas, VA, USA) were cultured in endothelial basal medium-2 complete medium.

SiSCR and siEGFR were obtained from Horizon Discovery. MiRNA negative control (miR-NC) and miR-218-5p mimics were obtained from Thermo Scientific (Rockford, IL, USA). The lentiviral vector alone and a vector containing miR-218-5p were from Open Biosystems (Huntsville, AL, USA), and EGFR ORF without 3′-UTR was from Vector Builders (Chicago, IL, USA). The empty control plasmid pcDNA3 was purchased from Addgene (Watertown, MA, USA). For transient transfection, JetPrime (Polyplus) reagents were used.

### 2.2. Immunoblotting Assay

Cells were lysed using RIPA buffer from Sigma-Aldrich (St. Louis, MO, USA) mixed with pierce protease and phosphatase inhibitors from Thermo Scientific (Rockford, IL). For each loading, 20–40 μg total protein was used and separated by 10–12% SDS-PAGE. Then, PVDF membrane and 5% nonfat milk were used to transfer and block the protein, respectively. The incubation period for primary antibodies against EGFR (1:1000, #4267, Cell Signaling Technology, USA), HIF-1α (1:1000, #610959, BD Transduction Laboratories™, San Diego, CA, USA), P-PKM2 (1:1000, #3827, Cell Signaling Technology, Danvers, MA, USA), PKM2 (1:1000, #4053, Cell Signaling Technology, Danvers, MA, USA), p-p65 (1:1000, #3033, Cell Signaling Technology, Danvers, MA, USA), p65 (1:1000, #8242, Cell Signaling Technology, Danvers, MA, USA), Phospho-p44/42 MAPK (Erk1/2) (1:1000, #4370, Cell Signaling Technology, Danvers, MA, USA), p44/42 MAPK (Erk1/2) (1:1000, #4695, Cell Signaling Technology, Danvers, MA, USA), GAPDH (AB_2617426, DSHB, Danvers, Ma, USA), β-actin (1:2000, #sc-47778, Santa Cruz Technology, Dallas, TX, USA), or α-tubulin (1:2000, #2125, Cell Signaling Technology, Danvers, MA, USA) was overnight at 4 °C in 5% bovine serum albumin (BSA, #9048-46-8, RPI Research Product, USA). IgG (H+L) goat anti-mouse (#Invitrogen 31430) and IgG (H+L) goat anti-rabbit (#Invitrogen 31460), and HRP (from Invitrogen™) were used as secondary antibodies. The incubation period for the secondary antibody was 2 h in 5% nonfat milk at room temperature. Chemiluminescence system reagents (Thermo Fisher Scientific, Waltham, MA, USA) and ChemiDoc Touch Imaging System were used to visualize and capture protein bands.

### 2.3. RNA Isolation and RT-qPCR

Trizol reagent and DNAse (Invitrogen) were used to extract total RNAs. For microRNA, TaqMan MicroRNA Assay Kit (ThermoScientific, Rockford, IL, USA) was used to detect the expressions of miR-218-5p and the internal control, U6 small nuclear RNA. For mRNA, the levels of KDR (VEGFR2), FLT4 (VEGFR3), and FLT1 (VEGFR1) expression were measured by the RT-qPCR method with SYBR-Green (Applied Biosystems, Carlsbad, CA, USA), and they were normalized by internal control β-actin. The following primers were used for qPCR:

KDR forward primer: 5′-AATGCTCAGCAGGATGGCAA-3′

KDR reverse primer: 5′-CCGGCTCTTTCGCTTACTGT-3′

FLT1 forward primer: 5′-TTCCGAAGCAAGGTGTGACT-3′

FLT1 reverse primer: 5′-CTCTCCTTCCGTCGGCATTT-3′

FLT4 forward primer: 5′-GAAGAGGAGGTCTGCATGGC-3′

FLT4 reverse primer: 5′-GTCTGTCTGGTTGTCCACAG-3′

β-actin forward primer: 5′-GACCTGACTGACTACCTCATGAAGAT-3′

β-actin reverse primer: 5′-GTCACACTTCATGATGGAGTTGAAGG-3′

### 2.4. Dual Luciferase Reporter Assay

Oligomers for wild-type (WT) and EGFR with 3′-UTR mutant with restriction sites for *Spe*I and *Hin*dIII were synthesized by IDT (Coralville, IA, USA). Oligomers were annealed and then subcloned into a pMIR-reporter Luciferase vector. The oligomers for EGFR 3′-UTR are given below, where the restriction sites are underlined. Italic fonts were used to show seed sequences. The 3′-UTR-mutant EGFR contains a 4-bp mutation, which is given in red letters in the seed sequence:

WT forward oligomer: 5′-CTAGTATGATGGA*AAGCACA*TTTAGCTTA-3′

WT reverse oligomer: 5′-AGCTTAAGCTAAA*TGTGCTT*TCCATCATA-3′

3′ UTR-mutant forward oligomer: 5′-CTAGTATGATGGA*AAGATGCA*TTTAGCTTA-3′

3′ UTR-mutant reverse oligomer: 5′-AGCTTAAGCTAAA*TGCATCT*TCCATCATA-3′

The 3′ UTR-mutant EGFR cannot bind to the miR-218-5p with the 4-bp mutation in seed sequence, which is used as a negative control. HEK293-T cells were transfected with miRNA negative control (miR-NC) or miR-218-5p mimics and pMIR-reporter (vector control) with WT or 3′ UTR-mutant EGFR using JetPrime reagents. Cells were collected after 48 h post-transfection, and Dual-Luciferase Reporter Assay System (Promega, Madison, WI, USA) was used to measure the Luciferase activity.

### 2.5. Cell Proliferation Assay

MiR-NC or miR-218-5p mimics were used to transfect As-T cells, and cells were seeded into 24-well plates. Cell proliferation was measured using a total number of live cells counted under a microscope.

### 2.6. Colony Formation Assay

Six-well plates were used for colony formation assay. The bottom of the wells was precoated with 0.5% SeaPlaque agarose containing DMEM complete medium. Then, transfected As-T cells with miR-NC or miR-218-5p mimics were suspended in 0.5% SPA at 3 × 10^3^/mL concentration and added on top of the precoated bottom layer. Colony formation was observed for 2 weeks in the incubator. To prevent desiccation, 0.5% SPA-containing medium was added every 3 to 4 days. Cell colonies were stained with a solution of nitro blue tetrazolium chloride overnight. The images were taken, and ImageJ Software version 1.53t was used to count the colonies.

### 2.7. Wound Healing Assay

The microRNAs transfected As-T cells were cultured into a six-well plate for 48 to 60 h. The wound on monolayer cell surfaces was made using a 200-μL sterile pipette tip when the cells were 90% confluence. The cells were washed, observed, and photographed at two time points (0 and 24 h). Migration distance was calculated using images at each time point.

### 2.8. Transwell Migration Assay

After transfection, the As-T cells diluted with DMEM medium containing 0.1% FBS were seeded into the upper chamber of a 24-well transwell plate with an 8.0-μm pore (Corning, Kennebunk, ME, USA). Medium containing 10% FBS as a chemoattractant source was added into the bottom chamber. Cells were incubated overnight at 37 °C, then washed by cold PBS. Cold 20% methanol and 3% formaldehyde were used to fix the cells. Finally, cells were stained with 0.1% crystal violet. Using a cotton swab, nonmigrated cells from the sides of the chamber were removed very carefully. The images were taken on the side of the bottom chamber using an inverted microscope. ImageJ software was used to count migrated cells plotted on a percentage scale.

### 2.9. Tube Formation Assay

Conditioned medium containing 0.2% FBS in DMEM was used for tube formation assay. The miR-NC and miR-218-5p were used to transfect As-T cells. When cells reached 95% confluence, the conditioned medium was collected and stored in a −80 °C freezer. EBM-2 basic medium containing 0.2% FBS was used to culture HUVECs for 24 h. Then, HUVECs were trypsinized, collected, and suspended with basic EBM-2 medium mixed with an equal amount of conditioned medium from each treatment group. The cell suspension was later introduced into the 96-well plate precoated with Matrigel and cultured for 6–18 h at 37 °C to observe tube formation. The images were taken using an inverted microscope. The total length of the tubes and tube numbers were measured using the ImageJ angiogenesis analyzer.

### 2.10. Lentivirus Packaging and Viral Transduction

MiR-NC and miR-218-5p in lentiviral vector plasmids were obtained from Open Biosystems (Huntsville, AL, USA). JetPrime was used for transfection. We generated lentiviral soup to overexpress miR-NC and miR-218-5p by co-transfecting 293T cells with a plasmid containing individual miR-NC or miR-218-5p, a PAX2 structural vector, and a pVSV-G envelope vector. The viral soup was obtained after 48 h, 60 h, and 72 h, and then filtered. Later, the soup was used to transduce As-T cells. MiR-NC and miR-218-5p overexpressing stable cells were selected by puromycin and verified under a microscope by observing RFP expression and the TaqMan RT-qPCR method.

### 2.11. In Vivo Tumor Growth Assay

Nude mice at 6 weeks were purchased from the Jackson Laboratory (Farmington, CT, USA). As-T cells stably overexpressing miR-NC or miR-218-5p at 2 × 10^6^ in 100 μL DMEM basal medium were subcutaneously injected into both flanks of the mice. Each treatment group had five mice receiving a total of ten injections. Tumor sizes were recorded every three to four days, and mice were sacrificed after three weeks of the implantation. Tumor volume was calculated using the formula of volume (mm^3^) = (width^2^ (mm^2^) × length (mm))/2. Tumor tissues were weighed, cut into halves, and stored at −80 °C for further detecting miRNA expression and immunoblotting assay. The animal studies were approved by the Institutional Animal Care and Use Committee (IACUC) at Thomas Jefferson University, and all the procedures were performed per the Guide for the Care and Use of Laboratory Animals.

### 2.12. Immunohistochemical Staining for CD31

Tumor tissues were fixed in Bouin’s solution. Then, the tissues were submitted to the Translational Research Lab at Sidney Kimmel Cancer Center, Thomas Jefferson University to embed the fixed tissues in the paraffin. CD31 antibody (1:100) was used for the staining, and streptavidin-biotin–horseradish peroxidase complex (SABC) formation was used for the detection of CD31 positive microvessels. The slides were observed under a low power (×10) microscope, and hot spots were found with the highest number of microvessels. The number of microvessels was counted under 20× magnification, and the mean number was presented as microvessel density and normalized by the control group.

### 2.13. Statistical Analysis

Three or more replicates were used for statistical analysis. All the results were shown as mean ± standard error. Student *t-*tests and analysis of variance (ANOVA) were used for comparing two groups and multiple groups, respectively. Note that *p* < 0.05 is considered statistically significant. All results were analyzed by GraphPad Prism and Excell.

## 3. Results

### 3.1. Upregulation of EGFR and Its Downstream Targets and Downregulation of miR-218-5p in As-T Cells

EGFR is a key oncogene in lung cancer to promote cancer initiation and development. We found that after exposure to the low dose of arsenic for 6 months, B2B cells underwent a malignant transformation to form colonies in vitro and to form tumors in nude mice [41,43]. As-T cells possessed a higher expression level of EGFR by 50% when compared to parental B2B cells (Figure 1A). EGFR upregulation was also confirmed in lung squamous cell carcinoma (LUSC, fold change of cancer vs. normal = 3.15, *p* < 0.001) and kidney renal clear cell carcinoma (KIRC, fold change of cancer vs. normal = 2.25, *p* < 0.001) using publicly available data from TCGA and following the relevant algorithms (http://starbase.sysu.edu.cn (accessed on 12 October 2022), Appendix A). As-T cells also showed activation of PKM2, ERK1/2, and p65 of NF-κB, as well as upregulation of HIF-1α (Figure 1B); silencing of the EGFR in As-T cells inhibited the phosphorylation of PKM2, ERK1/2, p65, and HIF-1α expression (Figure 1C), indicating that As exposure induced the expression of EGFR, which activated its downstream targets PKM2, ERK1/2, and p65, and promoted HIF-1α expression.

In order to determine the underlying mechanism of EGFR induction in As-T cells, we used the software TargetScan and miRDB to predict potential miRNAs that target EGFR, and miR-218-5p was among the top downregulated miRNAs. Furthermore, our previous miRNA sequencing results using parental B2B and As-T cells also showed significant downregulation of miR-218-5p. Consistent with the sequencing results, we found that the expression level of miR-218-5p was markedly suppressed in As-T cells to more than 100-fold (Figure 1D). The downregulation of miR-218-5p was observed in lung squamous cell carcinoma (LUSC, fold change of cancer vs. normal = 0.31, *p* < 0.001), kidney renal clear cell carcinoma (KIRC, fold change of cancer vs. normal = 0.81, *p* < 0.001), kidney renal papillary cell carcinoma (KIRP, fold change of cancer vs. normal = 0.74, *p* = 0.005), and bladder urothelial carcinoma (BLCA, fold change of cancer vs. normal = 0.65, *p* = 0.04) when compared to adjacent normal tissues (Appendix A). Meanwhile, miR-218-5p was decreased in both lung adenocarcinoma (LUAD) and LUSC in CancerMIRNome datasets. Lung cancer patients with higher expression of miR-218-5p showed better outcomes (Appendix A), suggesting that miR-218-5p may act as a tumor suppressor.

### 3.2. MiR-218-5p Functioned as a Tumor Suppressor in As-T Cells to Negatively Regulate Cell Proliferation, Colony Formation, Cell Migration, and Tube Formation In Vitro

Next, the functions of miR-218-5p in As-T cells were examined. We transiently transfected As-T cells with miR-NC mimics (negative control) or miR-218-5p mimics. We found that miR-218-5p overexpression suppressed cell proliferation (Figure 2A) and decreased the number of colonies (Figure 2B). Meanwhile, the wound healing and transwell migration were inhibited by miR-218-5p overexpression to approximately 40% and 25%, respectively (Figure 2C,D). Finally, the angiogenesis of HUVECs induced by As-T cells was dramatically suppressed by miR-218-5p overexpression, showing a decreased number of tubes to 45% and the total tube length to 60% compared to control (Figure 2E), demonstrating the antitumor role of miR-218-5p in As-T cells in vitro.

### 3.3. EGFR Was a Direct Target of miR-218-5p

We predicted that miR-218-5p might directly target EGFR by binding to its seed sequence at 3′-UTR using TargetScan software (Figure 3A). Next, we created a reporter construct containing wild-type 3′-UTR of EGFR (WT) and a mutant construct (EGFR 3′-UTR-mutant) by altering four bp marked in red (Figure 3B). Luciferase assay showed that overexpression of miR-218-5p significantly reduced wild-type reporter activity (Figure 3C). However, the mutant reporter activity did not change (Figure 3C), indicating that the transcriptional activation of EGFR was reduced due to the binding to the seed sequence of EGFR by miR-218-5p. Finally, overexpression of miR-218-5p in As-T cells attenuated EGFR expression to 40% in As-T cells (Figure 3D). In addition to As-T cells, we also observed that overexpression of miR-218-5p inhibited EGFR expression in chromium (Cr)-induced transformed cells (Cr-T) and skin cancer cell line A431 (Figure 3D), indicating that EGFR is a direct target of miR-218-5p.

### 3.4. MiR-218-5p Inhibited As-T Cells-Induced Tumor Growth and Angiogenesis and Decreased EGFR Expression In Vivo

Moreover, to study whether miR-218-5p inhibits As-T cell-induced tumor growth in vivo, we established stable As-T cell lines containing overexpressed miR-218-5p or miR-NC by lentiviral transduction and puromycin selection (Figure 4A). We found that the miR-218-5p overexpressed group had smaller tumor sizes compared to the miR-NC group (Figure 4B,C). At the end of the assay, 10 out of 10 implantations formed tumors in the miR-NC control group. However, 7 out of 10 implantations formed tumors in the miR-218-5p group (Table 1), indicating that forced expression of miR-218-5p in As-T cells significantly inhibited the initiation of tumorigenesis as well as tumor growth. Overexpression of miR-218-5p also decreased tumor weight (Figure 4D). An increased level of miR-218-5p in xenografts from As-T cells stably overexpressing the miR-218-5p group was also confirmed by TaqMan RT-PCR assay (Figure 4E). The expression of EGFR was reduced in tumor xenografts from the miR-218-5p expressing group (Figure 4F). Similarly, miR-218-5p overexpression significantly decreased the levels of VEGFRs, including FLT1 (VEGFR1), FLT4 (VEGFR3), and KDR (VEGFR2). In addition, the number of CD31-positive microvessels was reduced to 40% in the miR-218-5p overexpression group (Figure 4G). The results demonstrate that miR-218-5p acts as a tumor suppressor via targeting EGFR in As-T cells-induced tumor growth and angiogenesis in vivo.

### 3.5. EGFR Forced Expression Recovered miR-218-5p-Inhibited Cell Proliferation, Colony Formation, Migration, and Tube Formation

To determine whether miR-218-5p exerts its function through its target EGFR, we transiently overexpressed EGFR in As-T expressing miR-218-5p stable cells (Figure 5A) and then tested cancer-related biological processes. EGFR forced expression recovered miR-218-5p-suppressed cell proliferation (Figure 5A) and colony formation capability (Figure 5B). Moreover, forced expression of EGFR also restored cell migration ability by transwell migration assay (Figure 5C) and wound healing assay (Figure 5D), as well as increased tube formation (Figure 5E). These results suggest that miR-218-5p exhibits a tumor suppressor role in As-T cells through its target EGFR.

## 4. Discussion

Arsenic is a well-known carcinogen, and its exposure is closely related to lung cancer, especially NSCLC [44,45,46]. EGFR, also known as HER1/ERBB1, is a tyrosine kinase receptor of the human epidermal growth factor receptor (HER) family. It is a well-documented oncogene involved in cancer initiation, development, metastasis, and treatment resistance [47,48,49]. It has been reported that EGFR forms a functional network with VEGFRs and nAChRs to increase their activity [50,51], indicating the important role of EGFR in promoting angiogenesis and nicotine-induced malignant transformation. PKM2 is a multifunctional protein that primarily acts as pyruvate kinase and catalyzes the trans-phosphorylation from phosphoenolpyruvate to ADP for the last step of glycolysis to generate ATP. Besides its canonical role as pyruvate kinase, it also acts as a protein kinase or a protein coactivator [52,53,54]. Mounting evidence indicates that PKM2 is important for maintaining the malignant phenotype and tumor growth [52,55,56,57]. Previous studies have shown that EGFR induces PKM2 translocation to nuclei [58] through either ERK1/2-dependent phosphorylation [59] or a PKCε/NF-κB-dependent pathway to promote tumorigenesis [60]. PKM2 interacts with HIF-1α physically and acts as a coactivator for HIF-1α [61] to regulate its activity [62,63]. HIF-1α activation is important for tumorigenicity and angiogenesis in nude mice [64,65,66]; it is overexpressed in human cancers and is associated with angiogenesis induced by tumors and hypoxia [67,68]. It has been reported that EGFR activation is induced by arsenic exposure [20,21,22], and long-term exposure to arsenic at low doses results in increased EGFR expression due to inducing TGFα [11]. We also found that chronic arsenic exposure activates the EGFR/p-ERK/HIF-1α pathway [69]. Consistent with the studies above, we found that long-term exposure to arsenic dramatically upregulated EGFR expression levels, activated PKM2, ERK1/2, and p65 of the NF-κB subunit, and promoted HIF-1α expression, indicating that long-term exposure to arsenic induces the expression of EGFR and its downstream targets. However, the molecular mechanisms of EGFR induction by chronic arsenic exposure remain to be elucidated.

MicroRNAs (miRNAs) are small noncoding RNAs that repress mRNA expression or induce degradation through sequence-specific RNA–RNA interactions in the 3′ untranslated regions (3′-UTR) of targeted mRNA. MiRNAs participate in different biological regulatory events. Recent studies demonstrated that miR-218-5p served as a tumor suppressor to inhibit tumor growth and development by targeting endoplasmic reticulum oxidoreductase one alpha (ERO1A), COMMD8, CDK6, and EGFR in lung, liver, and gastric cancer [33,70,71,72]. It has been demonstrated that hypoxia and HIF-1α promote miR-218-5p expression [73,74], indicating that miR-218-5p is a hypoxia-inducible miRNA. Some miRNAs directly regulate EGFR to decrease its expression. For example, miR-7, miR-27a, and miR-128b negatively target EGFR and downregulate EGFR [75]. The alterations in mRNA expression exposed to different types of metals, including arsenic, have been studied [76,77,78]. EGFR is one of the direct targets of miR-218-5p [33,39]. However, the role and mechanism of miR-218-5p in both long-term and short-term arsenic exposure-induced carcinogenesis remain elusive.

In the present study, we found that miR-218-5p was dramatically downregulated in As-T cells, and miR-218-5p acted as a tumor suppressor to inhibit cell proliferation, migration, colony formation, tube formation in vitro, and As-T cells-induced tumor growth and angiogenesis in vivo, confirming the antitumor effect of miR-218-5p in arsenic-transformed cells. Furthermore, we demonstrated that miR-218-5p directly targeted EGFR. Finally, miR-218-5p exhibited its antitumor effect by inhibiting its direct target, EGFR and VEGFR receptors Flt-1, KDR, and FLT4. These findings are helpful to further understand the molecular mechanisms of arsenic-induced carcinogenesis and angiogenesis, and the miR-218-5p/EGFR signaling pathway may be a potential therapeutic target for the treatment of lung cancer induced by chronic arsenic exposure in the future.

## 5. Conclusions

Arsenic is a well-established group 1 human carcinogen that induces different types of cancers, including lung, bladder, skin, and liver cancers. It can affect various biological processes by DNA damage, reactive oxygen species (ROS) generation, abnormal epigenetic regulation, and inflammatory responses. To investigate the molecular mechanism underlying As-induced carcinogenesis and angiogenesis, we found that long-term As exposure markedly downregulated miR-218-5p expression and upregulated EGFR and its downstream targets. Moreover, miR-218-5p served as a tumor suppressor and directly targeted EGFR, which was required for miR-218-5p to suppress cell proliferation, colony formation, migration, angiogenesis, and tumor growth.

## Figures and Tables

**Figure 1 cancers-15-01204-f001:**
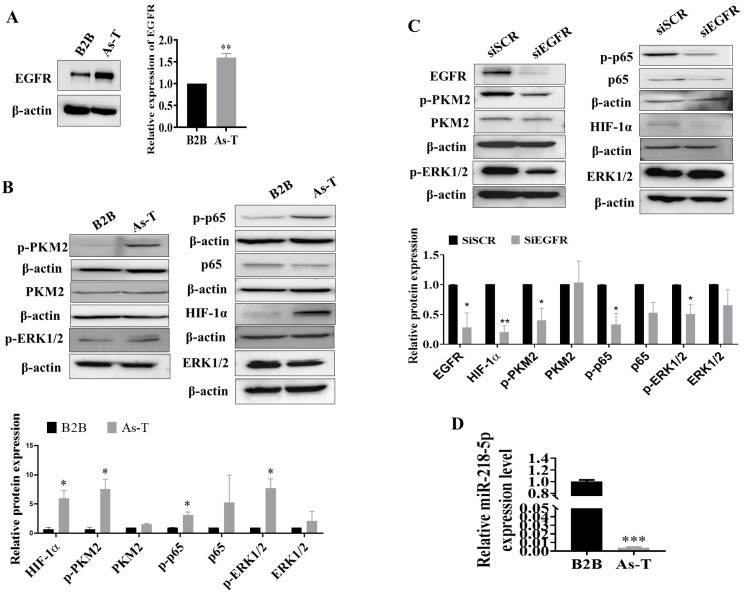
Upregulation of EGFR and its downstream targets PKM2, ERK1/2, NF-κB p65, and HIF-1α expression, and downregulation of miR-218-5p in As-T cells. (**A**) Expression of EGFR and β-actin in parental BEAS-2B (B2B) and As-T cells using an immunoblotting assay. (**B**) Expression levels of phospho-PKM2 (p-PKM2), PKM2, p-p65, p65, p-ERK1/2, ERK1/2, HIF-1α, and β-actin in protein extracts of B2B control and As-T cells. (**C**) As-T cells transfected with siRNA against EGFR or scrambled control (siSCR) at concentration 100 nM. After 72 h, immunoblotting assay was performed. (**D**) Expression of miR-218-5p in B2B and As-T cells using TaqMan real-time RT-PCR assay. Data represent the mean ± standard error (SE) of three independent experiments. * *p* < 0.05, ** *p* < 0.01, and *** *p* < 0.001 compared to B2B cells. The original western blots of this figure is Appendix A.

**Figure 2 cancers-15-01204-f002:**
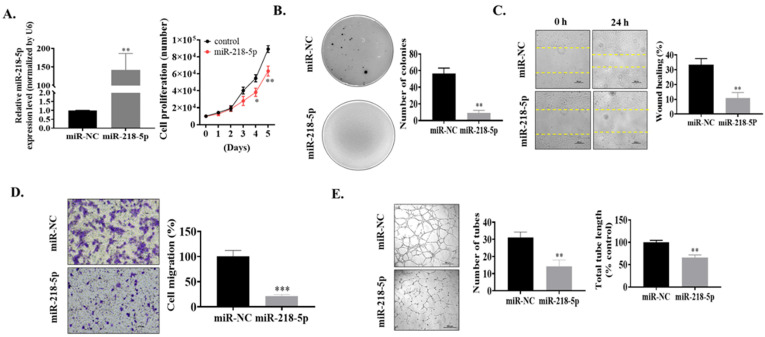
Tumor suppressor role of miR-218-5p in As-T cells. MiR-218-5p or miR-NC mimics were used to transfect As-T cells. After 24 h of transfection, the functional assays were performed. (**A**) Transfection efficiency confirmed by TaqMan real-time PCR. Cell proliferation assay was performed by counting the number of live cells using a cell counter under the microscope. (**B**) Soft agar colony formation assay. (**C**) Cell migration using wound healing assay. (**D**) Cell migration using transwell migration assay. Scale: 100 µm. (**E**) Endothelial cell tube formation assay. Data represent the mean ± SE of three independent experiments. * *p* < 0.05, ** *p* < 0.01, and *** *p* < 0.001 compared to miR-NC group. The original western blots of this figure is Appendix A.

**Figure 3 cancers-15-01204-f003:**
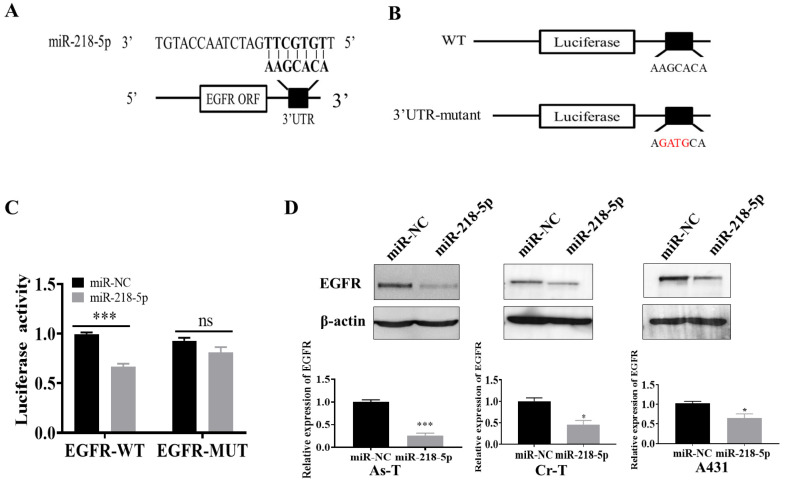
EGFR as a direct target of miR-218-5p. (**A**) The 3′-UTR region of EGFR to which miR-218-5p binds. (**B**) Luciferase reporter constructs containing wild-type (WT) and EGFR 3′-UTR mutant (MUT) plasmid. (**C**) Relative luciferase activities measured with miR-NC or miR-218-5p overexpression using HEK-293 T cells. Renilla luciferase reporter was used as an internal control. Data were represented by means ±SE. (**D**) As-T, Cr-T, and A431 cells transfected with miR-NC or miR-218-5p mimic, and the EGFR and β-actin protein expression measured using immunoblotting assay. * *p* < 0.05; *** *p* < 0.001; ns, not significant. The original western blots of this figure is Appendix A.

**Figure 4 cancers-15-01204-f004:**
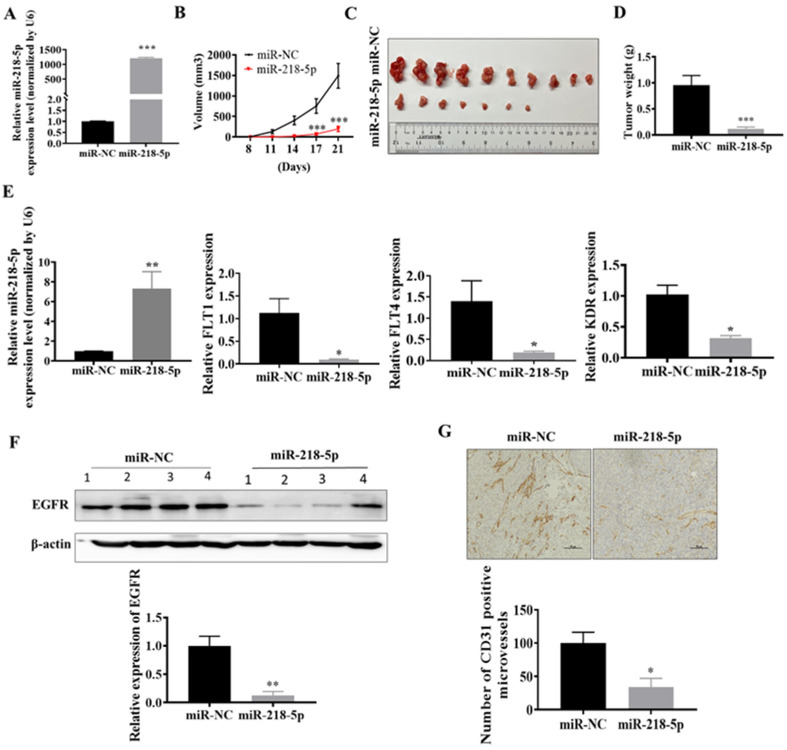
MiR-218-5p inhibited As-T cells-induced tumor growth and angiogenesis and decreased EGFR expression in vivo. (**A**) Stable As-T cell lines containing overexpressed miR-218-5p or miR-NC by transducing lentivirus and selecting puromycin. TaqMan real-time PCR was used to detect miR-218-5p expression. (**B**) As-T cells at 2 × 10^6^ overexpressing miR-NC or miR-218-5p subcutaneously injected into both flanks of nude mice (5 mice/group receiving 10 injections). Tumor size was measured every three or four days when tumors were visible. After 28 days of injection, mice were sacrificed, and tumor tissues were trimmed out. (**C**) Photograph exhibited xenografts obtained from the miR-NC and miR-218-5p groups. (**D**) Tumor weight from each group. (**E**) Expression of miR-218-5p, FLT1, FLT4, and KDR in the tumor tissues. (**F**) EGFR expression in tumor tissues from the miR-NC and miR-218-5p groups. (**G**) Number of CD31 positive microvessels by immunohistochemical staining assay, which was normalized by the miR-NC control group. Data were shown as means ± standard error. * *p* < 0.05, ** *p* < 0.01, *** *p* < 0.001 compared to the miR-NC group. The original western blots of this figure is Appendix A.

**Figure 5 cancers-15-01204-f005:**
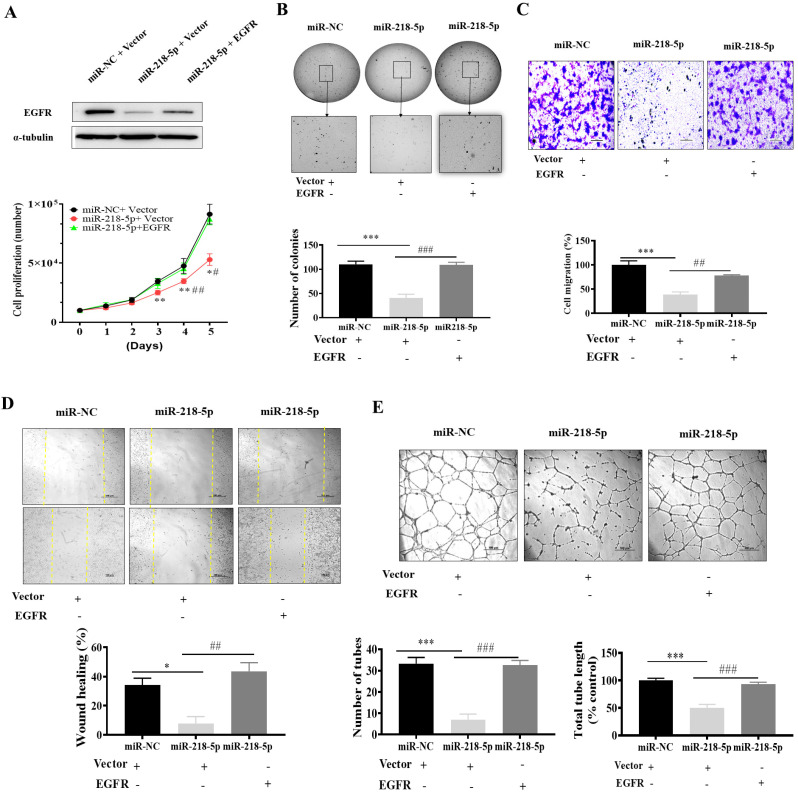
MiR-218-5p inhibited cell proliferation, colony formation, migration, and tube formation, which was recovered by forced expression of EGFR. As-T cells stably expressing miR-NC and miR-218-5p were transiently transfected by empty vector or vector carrying EGFR ORF. Immunoblotting assay was performed to detect EGFR and β-actin proteins after 72 h of post-transfection. (**A**) After 24 h of transfection, the cells were seeded into a 12-well plate, and cell proliferation assay was performed by manually counting live cells for 5 days. (**B**) Soft agar colony formation assay. (**C**) Cell migration using transwell assay. Scale: 100 µm. (**D**) Wound healing assay in the six-well plate. (**E**) HUVECs tube formation assay. Data were shown using line graphs and bar graphs from three independent experiments. The * denotes a significant difference between miR-NC+ empty vector and miR-218-5p + empty vector groups, and # represents a significant difference between miR-218-5p + empty vector and miR-218-5p + EGFR groups. * and # *p* < 0.05, ** and ## *p* < 0.01, *** and ### *p* < 0.001. The original western blots of this figure is Appendix A.

**Table 1 cancers-15-01204-t001:** Tumor formation in vivo.

Treatment Group	Number of Implantations	Number of Tumors
miR-NC	10	10
miR-218-5p	10	7

## Data Availability

The datasets generated/analyzed during the current study are available and will be provided upon request.

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
