# Peer review of "MiR-218-5p/EGFR Signaling in Arsenic-Induced Carcinogenesis"

_cancers, 2023, doi:10.3390/cancers15041204_

Round 1
Reviewer 1 Report (Previous Reviewer 3)
The goal of this research was to comprehend the mechanism behind the arsenic-induced overexpression of the EGFR in lung cancer. Results demonstrated that arsenic-induced transformed cells had enhanced EGFR expression and reduced miR-218-5p expression. The study found that miR-218-5p functions as a tumor suppressor and directly targets EGFR to prevent angiogenesis, colony formation, cell proliferation, and migration. According to the study's findings, the miR-218-5p/EGFR signaling pathway is critical for arsenic-induced carcinogenesis and angiogenesis. This understanding may be beneficial in developing future therapies for lung cancer brought on by chronic exposure to arsenic.
All of my comments and concerns have been addressed by the authors.
Reviewer 2 Report (Previous Reviewer 2)
The manuscript under revision was significantly improved and all my comments were answered adequately.
I suggest to accept the article for publication.
This manuscript is a resubmission of an earlier submission. The following is a list of the peer review reports and author responses from that submission.
Round 1
Reviewer 1 Report
The manuscript (cancers-2053234) by Islam R et al. reports the the role of miR-218-5p/EGFR signaling in arsenic-induced carcinogenesis. The study in general is well designed and performed. This reviewer has no major concerns. The minor concerns are some grammatic issues. Please do further proofread and correct minor language issues.
Reviewer 2 Report
The article under review is dedicated to the study of EGFR and miR-218-5p interplay in tumorigenesis in lung cells.
The article is well-written, easy to follow, the result’s presentation is good, data disclosure is fine (WB membranes and assay pics are provided), main conclusions are supported by the results.
Overall impression is good.
Some issues should be resolved before article acceptance:
1. You compared the expression of EGFR/miR-218-5p in lung squamous cell carcinoma, what about:
- lung adenocarcinoma (LUAD)
- skin cancers (SKCM and BCC)
- bladder (BLCA if I don’t mistake)
- liver (LIHC)
cancers, which you list in the introduction? Provide information in supplementary to strengthen the role of miR-218-5p in cancer control.
2. It is known that EGFR over expression is linked with decreased overall survival of lung adenocarcinoma patients, what about miR-218-5p? Please provide information about survival (Kaplan-Meier plots) in LUSC and LUAD cohorts.
3. Figure 1 b,c; Figure 5a – quantify WB in supplementary.
4. The action of miR-281-5p on EGFR should be confirmed using another cell line. My suggestion is A431 cells – skin adenocarcinoma with EGFR overexpression.
5. PECAM1 is good vascular marker but it is also interesting whether the As-induced transformation up-regulates the VEGFR expression in tumors because EGFR forms functional network with VEGFR and nAChRs. Could you clarify that issue?
6. The formation of functional network between VEGFR/nAChR (e.g. a7-nAChR or a4-nAChR)/EGFR is well known [10.1038/nrc3725], so activation of EGFR also can increase the activity of VEGFR or nAChR. Please discuss that facet of cell’s transformation, especially in the light of nicotine consumption, which can activate a7-nAChRs in addition to arsenic-induced EGFR activity.
7. Is it possible that HIF-1a can down-regulate miR-218-5p? Please discuss. Maybe that may help: 10.18632/oncotarget.22239. Also, I suppose that hypoxia can take place in tumor mass within your in vivo experiment, so some sentence in discussion will be useful.
Minor:
1. 2.2. Provide all antibody Cat# (including secondary) and clarify the buffer in which the incubation with primary and secondary Abs was carried out.
2. 2.3. Was your RNA treated by DNAse?
3. 2.11 The cells were injected subcutaneously?
4. 2.4 and 3.3. The phrase “mutant (MUT) EGFR” is confusing because EGFRVIII variant can also be named “mutant”, please change the “mutant” to “3’UTR mutant” and explain in the methods that “the 3’UTR-mutant EGFR binds miR-218-5p much worse than wt and was used as control” or something like that.
5. line 287-288 – it seems that reduction of wound healing was not 20% but 60% in comparison to mir-NC, please rephrase.
6. line 252 – it is better to write “inhibited phosphorylation of PKM-2” than “inhibited the expression of p-PKM2…”. The word expression is duplicated on 253 line.
How can you explain the bigger bands of phosphorylated protein bands in comparison to total proteins? Only antibody “effectiveness” or maybe that factors are generally activated in your cells?
7. Figure 4 C – please add the number of mice (tumors are arranged in pairs, right?).
8. Line 336 – Maybe “miR-218-5p-expressing” should be used instead of “miR-218-5p-inhibited”.
9. Line 63 – “arsenic exposure reduces EGFR degradation” – Arsenic actually does not cause EGFR degradation, while EGF down-regulates EGFR expression in cell lysates, please rephrase.
10. Some “micro-flaws”: check doi format in the references, provide the link in line 46 as reference, please highlight the seed sequences in UTRs in 2.4. in red to make them more visible, line 412 – “downregulate”, I suppose that “in the future” in line 18 is unnecessarily, line 431 “As is” is confusing, change to “Arsenic is”.
Generally, the manuscript under review is interesting and suitable for publication after listed issues will be resolved.
My suggestion: major revision.
Reviewer 3 Report
This study by Ranakul et al. demonstrated that miR-218-5p suppressed its key target, EGFR, to mediate its anticancer impact. This research also demonstrates the function of the miR-218-5p/EGFR signaling pathway in arsenic-induced angiogenesis and tumorigenesis, which, by focusing on the EGFR signaling pathway, may be effective for the future lung cancer treatment carried on by prolonged exposure to arsenic.
Comments-
1-In Figure 1, the Authors showed the EGFR and its signaling pathways in the cancer cells, but they haven't shown the MAPK expression, which is one of the main effectors of EGFR signaling.
2- In order to substantiate the outcomes of this study, it's also necessary to use another cell line in addition to the BEAS-B2 cells.
3- Please correct line 315 in the figure 3 legend "EGFR is the direct of miR-218-5p".
4- In addition, the authors also need to show the effect of miR-218-5p in EGFR down signaling molecules.